

# Spatial and temporal variations in surface snow chemistry along a traverse from Dome C toward South Pole in the framework of East Antarctic International Ice Sheet Traverse (EAIIST) project.

Simone Ventisette[1], Samuele Baldini[1], Claudio Artoni[2], Silvia Becagli[1,3], Laura Caiazzo[1,4], Barbara Delmonte[2], Massimo Frezzotti[5], Raffaello Nardin[1], Joel Savarino[6], Mirko Severi[1,3], Andrea Spolaor[3], Barbara Stenni[7], and Rita Traversi[1,3].

[1]Department of Chemistry "Ugo Schiff", University of Florence, Sesto Fiorentino, Florence I-50019, Italy
[2]Department of Environmental Science, University of Milano-Bicocca, Milan, Italy
[3]Institute of Polar Sciences, ISP-CNR, University of Venice, V. Torino 155, 30172 Venice-Mestre, Italy
[4]Laboratory for Observations and Measurements for Environment and Climate (SSPT-PROTER-OEM), ENEA C.R. Casaccia, 00123, Roma, Italy
[5]Department of Science, University of Roma Tre, Largo S. Leonardo Murialdo, 1, 00146, Roma, Italy.
[6]Institut des Géosciences de l'Environnement, Université Grenoble Alpes, CNRS, IRD, Grenoble INP, 38400 Grenoble, France
[7]Ca'Foscari University of Venice, Department of Environmental Sciences, Informatics and Statistics, Via Torino 155, 30172 Venice Mestre, Italy.

*Correspondence to*: Silvia Becagli (silvia.becagli@unifi.it)

**Abstract.** As part of the "East Antarctic International Ice Sheet Traverse" (EAIIST) project, 6 cm surface snow and snow pit samples (until about 2 m depth) were collected along a traverse from Dome C toward the geographic South Pole during the 2019-2020 Antarctic campaign. Results on spatial distribution of major ions are here reported to understand deposition and post deposition processes in sites with very low snow accumulation rate in the East Antarctic Plateau where megadune and wind crust areas are present.

The volcanic signature of Pinatubo eruption (occurred in 1991) was clearly visible in the non-sea salt $SO_4^{2-}$ stratigraphy from two snow pits (AGO-5 and PALEO) allowing the determination of annual accumulation rates that revealed to be 25.7 and 22.6 mm of water equivalent/year, respectively at the two sites. Moreover, a decreasing trend in accumulation rate as the distance from the Indian Ocean increases was detected. Mineral dust concentration and size show presence of a criptotephra layer in AGO5 and PALEO stratigraphies which is stratigraphically compatible with the deposition of volcanic ash related to the Puyehue-Cordón Caulle explosive eruption occurred in June 2011. The ssNa$^+$ fraction, accounting for the 92.5% of the total Na$^+$, is preserved stably in the snow layers and was chosen as marker of sea spray deposition. Despite the very low accumulation rate in this area, the main deposition process of sea spray aerosol is the wet deposition. Conversely, both biogenic and crustal nssSO$_4^{2-}$ are dry deposited, the total flux of nssSO$_4^{2-}$ resulted to be constant in the Antarctic plateau, but the biogenic to crustal ratio increases as distance from Dome C increases. The presence and quantification (by nssCa2+) of a dry deposited crustal source, as the biogenic one, sheds light on the interpretation of nss $SO_4^{2-}$ biogenic stratigraphy during glacial and interglacial time in Antarctic ice cores. NssCl$^-$ represent the fraction of Cl$^-$ deposited as HCl and arises from the exchange



reactions between chloride in the sea salt aerosol and acidic species such as $H_2SO_4$ and $HNO_3$ that occurs both into the
atmosphere (in this case HCl is deposited by wet deposition) and into the snow (at the expenses of NaCl or $MgCl_2$ deposited
as sea salt aerosol). The latter process could be particularly efficient in sites affected by wind crust formation, probably because
of a longer exposure time of the snow layers to the atmosphere favouring the HCl volatilization. Another important marker in
ice core is HNO3, that in the considered sites is found at very high concentration in the most superficial 3 cm of snow due to

the uptake by superficial snow and possibly concentration effects from the layers beneath, but it is reversibly deposited. The
depth of the active layer for $HNO_3$ reemission was calculated and it spans from 22 cm to 12 cm; in addition, the concentration
preserved in the snow decreases as the accumulation rate decreases, but wind scouring increases the efficiency of re-emission
processes in the active layer. The knowledge and quantification of all the above reported processes will allow the interpretation
of the ice core stratigraphies in low accumulation site likely hopefully recording, at selected sites, the climate history of more

than one million years ago.

## 1 Intruduction

Understanding the spatial and temporal variability of snow chemical composition is a prerequisite in the glaciological and
glaciochemical studies. It allows obtaining reliable information about past variations in the chemical composition of snow
recorded in ice cores, and to evaluate the mass transport of chemical markers, which, due to the high spatial variability in

aerosol composition, scavenging processes, snow accumulation and wind distribution, are fundamental to characterize
climatically unexplored area of the Antarctic ice sheet (Becagli et al., 2005; Bertler et al., 2005; Benassai et al., 2005).
Several studies investigated the spatial variation of glaciochemical data across the ice sheet along traverses during the
International Trans-Antarctic Scientific Expedition (ITASE), e.g., Dumont d'Urville Station (DDU) to Dome C, coast-interior
traverse in Adélie Land, Syowa Station to Dome F, Terra Nova Bay to Dome C, Zhongshan Station to Dome A and US ITASE

in West Antarctica (Dahe et al., 1992; Dixon et al. 2013; Legrand and Delmas 1985; Mulvaney and Wolff 1994; Proposito et
al. 2002; Shi et al. 2021; Suzuki et al. 2002), but a large part of the Antarctic ice sheet is still unexplored.
In this framework, the EAIIST project (East Antarctic International Ice Sheet Traverse) is an international collaboration aiming
to study the interior of the Antarctic Plateau between the Italian-French "Concordia" station (75° 06' 01" S, 123° 20' 48" E)
and the geographic South Pole. The general objective of the project is to study the Antarctic ice sheet in its most arid and

unknown places, which presents unique and extraordinary surface morphological characteristics. Indeed, the sampled area has
a high orographic interest containing several unique geographical features such as "megadune" or "wind crusts" area,
accompanied by physical and chemical processes of the snowpack such as post-depositional variations that are not yet fully
understood. Besides, despite the formation of wind crusts had been accurately studied by field observations and wind tunnel
experiments (e.g. Sommer et al., 2018), its effect on surface mass balance, isotopic and chemical composition is still poorly

understood.



This study presents chemistry data from 6 cm surface snow samples and about 2 m depth snow pits collected along the EAIIST Concordia Station to South Pole 2019-20 traverse. We use these data to determine the spatial variability of chemical deposition and post deposition processes over an inaccessible area of the Antarctic continent. Besides, as this area is characterized by extremely low accumulation rate and extensively recrystallized snow, we use the chemical variation as function of space and depth to further infer surface and sub-surface conditions along the traverse route.

## 2 The studied area

The surface snow and snow-pit samples were collected during the 2019-2020 Antarctic campaign, in an area of the East Antarctic Plateau that was crossed by the scientific traverse carried out in the framework of the EAIIST project (Fig. 1).

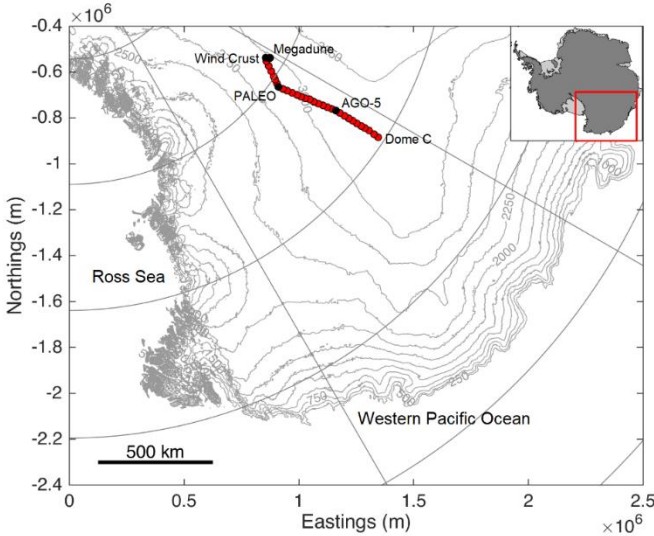

**Figure 1.** Map of Antarctica with the sampling sites. Superficial snow sampling sites of the EAIIST traverse are reported in red. The sites in which snow pit samples were collected are reported in black. Dome C, where Concordia Station is located, is also reported on the map as a reference point.

The route started from Dome C and closely followed the 123°E Meridian with small variations in altitude, which remains close to 3000 m a.s.l. (3061±116m) for the whole traverse (Fig. 2a and Table 1S) to minimize the variability in the composition of the snow due to the altitude variations. It has been shown, in fact, that the concentrations of dissolved components in the snow change in relation to several intrinsic parameters of the sampling site such as the distance from the sea, the altitude and the accumulation rate (Becagli et al. 2004; Bertler et al. 2005; Khodzher et al. 2014; Proposito et al. 2002; Udisti et al. 2004).

The distance from the Indian Ocean increases as the distance from Dome C increases, conversely the distance from Ross Sea decreases making quite constant the distance from the nearest ocean. Indeed, previous studies shown that the main part of the



sampled sites mainly receive air masses from the Indian Ocean and only the few farthest sites from Dome C are affected also by air masses coming from the Ross Sea (Scarchilli et al., 2011; Sodemann and Stohl, 2009).

The traverse crosses areas of the ice sheet characterized by the presence of wind crust and megadunes. Megadune fields occupy large areas in the interior of the East Antarctic ice sheet and are the result of peculiar snow accumulation and redistribution

processes. These areas are usually characterized by slightly steeper regional slope and the presence of highly persistent katabatic winds. Megadunes are characterized by an amplitude of 2 to 4 m with a wavelength ranging from 2 to 5 km. Their net accumulation differs from the adjacent non-megadune areas from 25% (leeward or erosion faces) to 120% (windward or accumulation faces). Erosion faces are characterized by glazed, sastrugi-free surfaces and extensive hoar formation whilst accumulation faces are covered by large rough sastrugi up to 1.5 m in height (Frezzotti et al., 2002; Traversa et al., 2023).

At five sampling sites, named AGO-5, PALEO, Wind Crust (WCru), Megadune Erosion (MGD-E), and Megadune Accumulation (MGD-A), additional snow-pits sampling was conducted.

The list of sampling sites together with geographical characteristics and the type of sampling performed are reported in Table 1S.

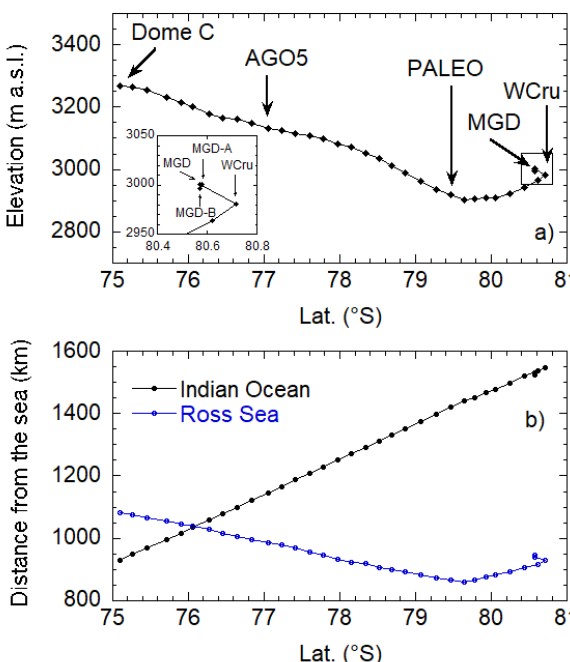

**Figure 2.** Elevation (plot a) and distance from the sea (plot b) respect to the latitude for each sampling sites. The sampling points where snow pits were dug (AGO5, PALEO, WCru, MGD-A and MGD-E) are reported in plot a. The detail of MGDs and WCru area is reported as enlarged square inside plot a.



## 3 Methods

### 3.1. Sampling procedures


All samples were collected using 50 mL polyethylene Corning® tubes by personnel wearing low particulate release clothing, polyethylene gloves (over silk thermal gloves) to minimize external contamination. Surface snow samples (uppermost 6 cm of snow) were collected by directly inserting the tubes into the surface snow of the sampling site (upwind and at a sufficient distance from the convoy to reduce the contamination risk). The density of the surface snow was determined by inserting a

stainless-steel cylinder open on two sides in the snow; the recovered (known) volume of snow was stored in a plastic bag and weighed in the convoy's cold laboratory.

In the above mentioned 5 sites about 2 m snow-pit were dig and samples were collected using the following procedure: first, the surface wall of the pit was decontaminated after the excavation by removing approximately 20 cm of the vertical layer with a plastic scraper. Subsequently, Corning® tubes were inserted one below the other into the vertical wall, collecting samples at

a resolution of 30 mm (the diameter of the tube).

After sampling, the tubes were put in their rack, sealed in polyethylene bag, and kept at -20°C throughout transport and up to the time of analysis. Any interaction with the samples was performed within a clean room (Class 10000), wearing low particle release clothing. Samples were left thawing at room temperature just before the analysis inside a class 100 laminar flow hood, in the same tube in which they were collected.


### 3.2 Analysis

For each sample, the cationic ($Na^+$, $NH_4^+$, $K^+$, $Mg^{2+}$, $Ca^{2+}$) and anionic ($Cl^-$, $NO_3^-$, $SO_4^{2-}$) content was determined by Ion Chromatography using an integrated system of two ion chromatographs working in parallel. A new method was optimized and

applied to these samples allowing to obtain fast (5.50 minutes) and high sensitivity measurements without compromising resolution between chromatographic peaks. The risk of contamination during the sample loading was minimized by using an autosampler (222XL Liquid Handler, Gilson, Middleton, WI, USA) with a single steel needle connected to a peristaltic pump (Gilson Minipuls 3) remotely controlled (Morganti et al., 2007).

For anions determination, samples were loaded into a ThermoFisher - Dionex Ion Pac TAC-2 (3 x 35 mm) by a peristaltic

pump with an average flow rate of 1.30 mL/min for 0.75 minutes, corresponding to 0.975 mL of sample loaded into the TAC. A ThermoFisher Dionex DX500 ion chromatograph (Sunnyvale, CA, USA) equipped with a Dionex Ion Pac AS4A 4 x 25 mm (10-32) analytical column and a Dionex AERS 500 self-regenerating conductivity suppressor (4 mm) was used for anions determination. Chromatographic peaks were detected by a Dionex CD25 conductivity detector, which uses the eluent stream leaving the detector as a regenerating solution. An isocratic run was performed using a 2.0 mM/3.0 mM NaHCO3/Na2CO3 as



eluent with a 2.0 mL/min flow. In these condition organic anions (Acetate, Formate, and Methanesulfonate) are not detectable as they lie in the water dip, but they are separate from Chloride peak and do not affect its determination.

The determination of the cationic content was obtained using a Dionex ICS 1000 ThermoFisher ion chromatograph equipped with a Dionex CERS 500 self-regenerating conductivity suppressor (4 mm, suppression current 180 mA) and a Dionex DS6 conductivity detector. Thanks to the low concentration of the ions and the absence of other ions interference, separation was

obtained using two Dionex CG12 guard columns (4 x 50 mm) that allow obtaining an excellent peak resolution in a shorter total run time than using a single separation column. Isocratic run with 12.5 mN $H_2SO_4$ sample injection as eluent at flow, 2.0 mL/min was used. Besides, the low retention time for all the ions allow obtaining high and narrow peaks, therefore increasing the sensitivity of the measurements that are obtained by the injection of only 200 μL of sample.

Dionex Chromeleon® chromatography software was used for instrumentation control and data acquisition.

External 5-point calibration curves were used for quantitative determination of each ion, standard solutions for calibrations were prepared daily diluting 1000 mg/L Merck standard solutions (Darmstadt, Germany) with MilliQ ultrapure water (Resistivity > 18 MΩ·cm).

Detection limits (d.l.) and reproducibility are reported in Table 1.

**Table 1**. Detection limit and reproducibility (%) for each ion. Detection limits are obtained as the concentration corresponding to a signal of three times the standard deviation of 10 repetitions of a 10 μg/L standard solution.

| Ion | d.l. (mg/L) | Repr. (%) |
|---|---|---|
| $Cl^-$ | 4.9 | 3.4 |
| $NO_3^-$ | 3.0 | 2.7 |
| $SO_4^{2-}$ | 2.1 | 1.3 |
| $Na^+$ | 0.1 | 1.9 |
| $NH_4^+$ | 0.6 | 3.8 |
| $K^+$ | 0.3 | 2.9 |
| $Mg^{2+}$ | 0.2 | 1.6 |
| $Ca^{2+}$ | 0.2 | 1.9 |

Blanks were evaluated before and after each calibration procedure resulting below the detection limit for each determined ion. Within the same snow pits dug for chemical sampling, a parallel line of samples was dedicated to mineral dust samples. These

were analyzed at the EUROCOLD Laboratory of Milano-Bicocca in clean room ISO6 by means of a Beckman Coulter Multisizer 4e equipped with a 30 μm orifice tube, following the standard analytical protocol for dust analyses in ice cores (Delmonte et al., 2004).



## 4. Results and discussion

### 4.1 Sea-salt and non-sea-salt fractions

It is well known that $Na^+$ and $Ca^{2+}$ have both sea spray and crustal sources, therefore sea-salt (ss) and non-sea-salt (nss) fraction of both ions was calculated using a simple two-equations system (Röthlisberger et al. 2002; Udisti et al. 2012)

$$ssNa^+ = Na^+ - 0.562*nssCa^{2+} \qquad \text{(equation 1)}$$
$$nssCa^{2+} = Ca^{2+} - 0.038*ssNa^+ \qquad \text{(equation 2)}$$


where $0.562 = Na^+/Ca^{2+}$ (w/w) in the crust (Bowen 1979) and $0.038 = Ca^{2+}/Na^+$ (w/w) in seawater (Bowen 1979).

The sea salt represents the dominant contribution to $Na^+$ concentration, accounting for the 92.5% of the total Na budget measured in superficial snow in all the sampled sites in this area. $SsNa^+$ concentration was chosen to assess the contribution of sea salt aerosol for the other ions present in the sea water ($Mg^{2+}$, $K^+$, $Ca^{2+}$ and $Cl^-$). The correlation plots of these ions respect

to $ssNa^+$ show for $Mg^{2+}$ and $K^+$ a dominant sea salt source demonstrated also by a good correlation with $ssNa^+$ ($R^2 > 0.8$) and ratios with $ssNa^+$ close to the sea water ratios (Fig. 1Sa e b). Due to $Ca^{2+}$ crustal source, for this ion a worse correlation than $K^+$ and $Mg^{2+}$, and a slope of the regression line higher than ($Ca^{2+}/Na^+$) sea water ratio was found (Fig.1Sc). Indeed, by the equation 1 and 2 is possible to estimate the $nssCa^{2+}$ contribution that as average represents the main contribution (77.6%) of total calcium budget.

In the case of $Cl^-$, a correlation with $ssNa^+$ was found, but all (except one) samples have a $Cl^-/Na^+$ ratio higher than sea water one. This suggests the presence of an alternative source for this ion.

HCl can arise by the interaction between sea spray particles (containing NaCl and $MgCl_2$) and some acidic species (mainly $HNO_3$ and $H_2SO_4$; reaction 1 and 2 respectively). The volcanic source of HCl can be excluded due to its sporadic and extreme character.


$$2NaCl \text{ (or } MgCl_2) + H_2SO_4 \rightarrow Na_2SO_4 \text{ (or } MgCl_2) + 2 HCl \qquad \text{(reaction 1)}$$
$$NaCl \text{ (or } \frac{1}{2} MgCl_2) + HNO_3 \rightarrow NaNO_3 \text{ (or } \frac{1}{2} Mg(NO_3)_2 ) + HCl \qquad \text{(reaction 2)}$$

These acid-base exchange reactions produce gaseous HCl, which follows different transport and deposition pathways than the sea-spray particles (Kerminen et al., 2000; McInnes et al. 1994). The amount of $Cl^-$ present in snow layer as HCl ($nssCl^-$ or $excCl^-$) can be calculated using the following equation:

$$nssCl^- = Cl^- - ssNa^+ *(Cl^-/Na^+)_{sw} \qquad \text{(equation 4)}$$






where $Cl^-$ and $ssNa^+$ are the total $Cl^-$ and sea-salt $Na^+$ concentration. $(Cl^-/Na^+)_{sw}$ is the $Cl^-$ to $Na^+$ ratio in sea water which is 1.81 w/w (Bowen 1979). Using this equation is possible to highlight that in superficial snow the fraction of $Cl^-$ present as HCl is on average the 46% .

The $ssNa^+$ was also used to calculate the non-sea salt sulphate fraction ($nssSO_4^{2-}$) using the following equation:


$$nssSO_4^{2-} = tot\ SO_4^{2-} - 0.25* ssNa^+ \hspace{4cm} \text{(equation 5)}$$

where 0.25 is the $SO_4^{2-}/Na^+$ ratio in sea water (Bowen 1979). The value of $SO_4^{2-}/Na^+$ in sea water was used instead of the $SO_4^{2-}/Na^+$ ratio of 0.09 (w/w) characteristic of sea-salt aerosol arising from frost flowers (Legrand et al., 2017) as the latter source

was demonstrated to be significant only for winter samples and has a minor impact respect to the open ocean source to the annual budget (Legrand et al., 2017; Udisti et al., 2012). Anyway, the impact of sea salt $SO_4^{2-}$ is very low in this area, accounting as average only for 12.4% of total sulphate. This value agrees with previous data obtained in the Antarctic Plateau (Khodzher et al. 2014; Li et al. 2016; Shi et al. 2021; Traversi et al. 2004; Uemura et al. 2016), showing that the the main source of $SO_4^{2-}$ is from secondary sources. Previous studies by sulfur isotopic measurements show that the contribution of

biogenic source (from oxidation of phytoplanktonic dimethylsulphide) in Antarctica is largely dominant over the other sources of sulfate (Khodzher et al. 2014; Uemura et al. 2016). The volcanic source of $nssSO_4^{2-}$ (from oxidation of volcanic $SO_2$) is sporadic, in correspondence of volcanic eruption and it is well evident in central Antarctic Plateau as concentration spikes of $nssSO_4^{2-}$ superimposed to the biogenic background (Castellano et al. 2005). The presence of sulfate spikes ascribable to historical know volcanic eruptions allows obtaining the dating of the snow layers and to assess the accumulation rate.


**4.2 Volcanic stratigraphy and accumulation rate trend**

Due to the very low accumulation rate in all the sampled sites the stratigraphic dating of snow layer is unpracticable. Therefore, the identification of volcanic signatures in the snow pit sulfate stratigraphies is used to calculate the accumulation rate. Since

volcanic eruptions emit (amongst other components) large amount of $SO_2$ in the atmosphere, if the volcanic eruption is sufficiently strong, $SO_2$ is able to reach the stratosphere and it can have hemispheric distribution. Therefore, it is possible to identify past volcanic eruptions in snow layers occurred at least in Southern hemisphere (but also until 20°N), as concentration spikes of $nssSO_4^{2-}$ arising from the stratospheric oxidation of $SO_2$ and subsequently deposition on the Antarctic plateau. These concentration spikes are superimposed to the biogenic background and can be statistically recognizable from the background

values (Castellano et al., 2005). By matching these volcanic signatures with historical records or other already dated cores, it is possible to date ice records (Castellano et al., 2005; Severi et al., 2007; Nardin et al., 2020).

To highlight the volcanic source an iterative method from Castellano et al. (2005) was adopted, consisting in calculating a $nssSO_4^{2-}$ background from biogenic source and looking for at least two consecutive samples with $nssSO_4^{2-}$ concentration overcoming the background $+ 2\sigma$. The volcanic eruption of the Pinatubo occurred in 1991 CE and recorded in Antarctic snow



layer in 1992 CE deposition (Cole-Dai and Mosley-Thompson, 1999) can be found in the AGO-5 and PALEO records at the

depths 199.5 cm and 172.5 cm respectively (Fig. 3). By taking into account the snow density of the snow pits layers, the mean

accumulation rates for the 1992-2019 period at the two sites were estimated to be 25.7 and 22.6 mm of water equivalent/year

respectively.

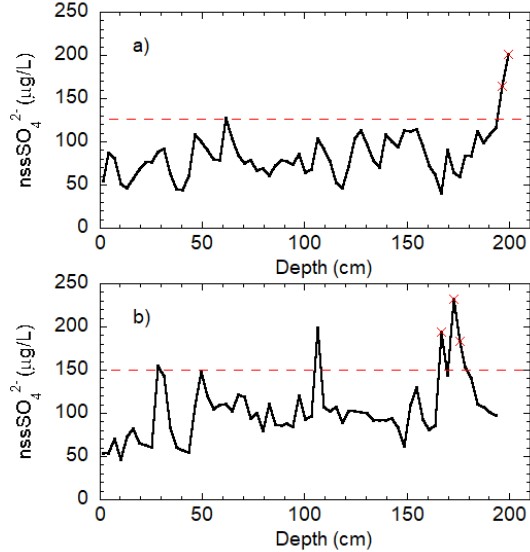

**Figure 3**. NssSO$_4^{2-}$ stratigraphies for AGO-5 (a) and PALEO (b) snow pits. The volcanic threshold obtained by the iterative

method is shown as a red dashed horizontal line. Sample points ascribed to a volcanic signature are highlighted as red crosses.

To increase the reliability of accumulation rate obtained by the only signature of Pinatubo eruption the insoluble particle

stratigraphy obtained at AGO5 and PALEO were compared to find dust events recorded at both sites.

The concentration profiles of insoluble particles obtained for AGO5 and PALEO snow pits show background dust

concentration levels around 15 ppb at both sites. Superimposed to background, AGO5 shows a 18 cm-broad peak in the

mineral dust concentration record from about 55.5 cm depth to 73.5 cm depth (Fig. 4), when concentration levels rise to ca. 4

times above background; similarly, the PALEO snowpit stratigraphy shows a 15-cm wide peak between 34.5 and 49.5 cm

depth, when concentrations rise to ~5 times background levels, on average. Within both peaks it is possible to observe that

mineral dust grain size progressively decreases since the beginning of the event, or initial dust rise, until a minimum centred

around 61.5-64.5 cm depth at AGO5 and 37.5-40.5 cm at PALEO, when the modal values of the dust grain size distribution

decrease below the typical background levels of ca. 2 µm. The Weibull functions used to fit dust spectral data of volume-size

distributions at both sites show modal values as low as 1.73 (±0.02) µm for AGO5 and 1.70 (±0.02) µm for PALEO (Fig 5) at

the time of dust size minima occurring when the Coarse Particle Percentage (CPP%) drops to -10% with respect to background

(arrows in Fig.4). Both events are associated with an almost concomitant increase in nssSO$_4^{2-}$ concentration even if the nssSO$_4^{2-}$

values lie in the variability range of biogenic background. All these peculiar features are typical of cryptotephra layers (Narcisi



et al., 2019; Delmonte et al., 2004) occasionally found in Antarctic snow. Therefore, we believe that these two peaks represent the same volcanic event preserved in the snowpack, which can be used as additional relative chronostratigraphic marker between the two snow pits. The stratigraphic location of the peak is compatible with deposition of volcanic ash from the

Puyehue-Cordón Caulle (Chile) explosive eruption event of June 2011 detected also in West Antarctica (Koffmann et al., 2017), although glass geochemistry is necessary to unequivocally infer the volcanic source.

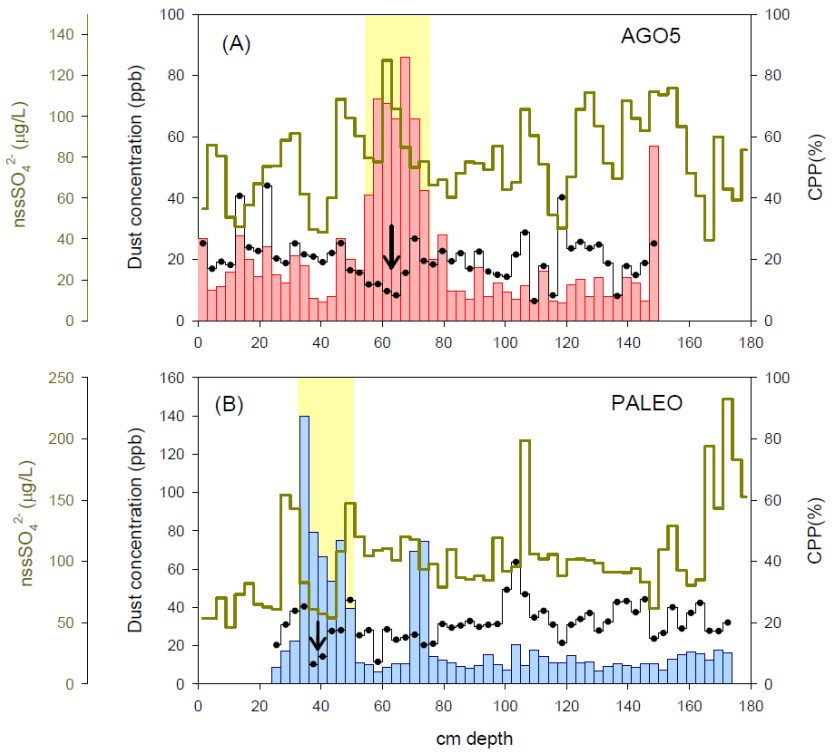

**Figure 4**. Dust concentration profile for the AGO5 (plot A) and PALEO (plot B) snow pits (ppb, or ng of dust per mL of

sample), with the nssSO$_4^{2-}$ profile and the Coarse Particle Percentage (CPP, %) index (Delmonte et al., 2004) on the right axis. The black arrow indicates the dust size minimum. NssSO$_4^{2-}$ concentrations are shown by the green step line, dust concentrations by pink columns and CPP (%) by black circles and interpolating black line. Yellow shadowed areas mark dust concentration peaks.




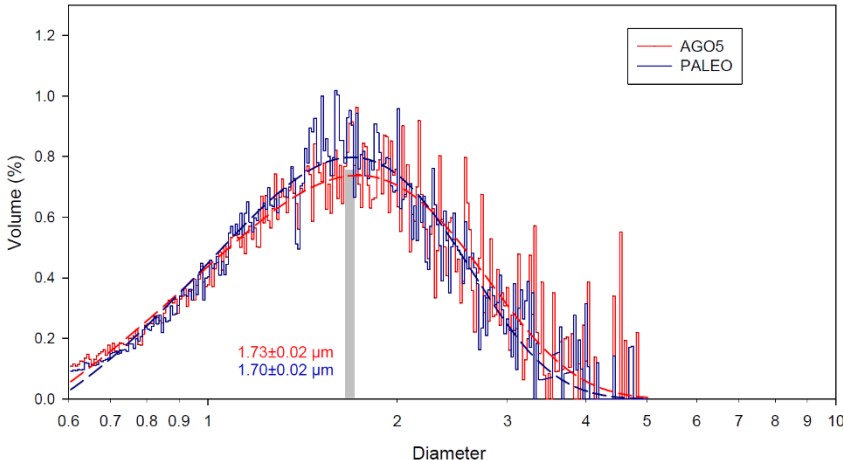


**Figure 5**: Dust volume (mass)-size distribution for AGO5 and PALEO samples corresponding to the layers of dust size minima indicated by arrows in figure 4. For both spectra, the dust mode has been calculated through a 4-parameter Weibull distribution used to fit data. Modal values are almost identical, i.e. 1.73 (±0.02) µm for AGO5 and 1.70 (±0.02) µm for PALEO site.

Comparing the values of accumulation rate here assessed with the values estimated at Dome C obtained by previous studies (Traversi et al.,2009; Caiazzo et al., 2021), it is possible to highlight a decreasing trend of accumulation rate as the distance from the Indian Ocean increases (Table 2).

**4.3 Deposition and post deposition processes as function of accumulation rate and glaciological features.**

To understand the deposition and re-emission processes of the analyzed ions is fundamental the study of the concentration and

deposition flux trend as function of the sampling site locations that, as above described, differ mainly for accumulation rate, snow redistribution by the winds and eventual wind crust formation.

By using a linear regression to fit the decreasing trend of snow accumulation as distance from the Indian Ocean increases, it was possible to evaluate the accumulation rate for each site and therefore the deposition flux of each ion by the equation:

$$F = C * Acc$$

Where F is the deposition flux (µg/m² yr), C is the ion concentration (µg/L) and Acc is the accumulation rate (mm w.e./yr).

Fig. 6 reports the concentrations and fluxes of $ssNa^+$, $nssCa^{2+}$, $nssSO_4^{2-}$, $nssCl^-$ and $NO_3^-$ in surface snow samples and concentrations mean ± 1σ in the snow pits. These markers are selected to study the spatial variability and deposition processes

of the different types of aerosols ($ssNa^+$ for sea salt aerosol, $nssCa^{2+}$ for crustal aerosol, $nssSO_4^{2-}$ for biogenic aerosol) and species mainly present as gas (HCl and $HNO_3$) present into the Antarctic atmosphere.

**Table 2.** Accumulation rates at different sites along the EAIIST traverse





| Sampling point | Dist. Indian Ocean (km) | Years of accumulation | Acc. Rate (mm snow/yr) | Acc. Rate (mm we/yr) | References |
|---|---|---|---|---|---|
| Dome C | 930 | 1964-1992 | 80-88 | 32.8 | (Traversi et al., 2009) |
| Dome C | 930 | 1992-2016 | 90 | 35.3 | (Caiazzo et al., 2021) |
| AGO-5 | 1165 | 1992-2019 | 73.9 | 25.7 | This study |
| PALEO | 1440 | 1992-2019 | 63.9 | 22.6 | This study |



Regarding the comparison between range of concentrations in the snow pit respect to those at the surface two different patterns are visible: (i) ions presenting concentrations in the superficial layer in the range of mean $\pm 1\sigma$ in the snow pits; these are ssNa$^+$, nssCa$^{2+}$ and nssSO$_4^{2-}$ which are considered stable in the snow layers and (ii) ions presenting concentration in the superficial snow samples higher than mean $+ \sigma$ in the snow pits, namely Cl$^-$ (for the sites showing the lowest accumulation rates) and NO$_3^-$, that are considered reversely deposited species.

The concentration pattern of ssNa$^+$ (and therefore also for the other ions mainly arising from sea spray: K$^+$ and Mg$^{2+}$) shows a constant trend in concentrations and a decreasing trend in fluxes as latitude, distance from Dome C and distance to the Indian Ocean increase. Besides, ssNa$^+$ concentrations measured in this area are in the range of that measured at Dome C (Udisti et al., 2004), Dome Fuji (Iizuka et al., 2006) and in general in the Antarctic Plateau (Bertler et al., 2005).

This different pattern of concentration and fluxes is typical of wet deposited species in low accumulation sites. In this area, the snowflakes contain the same amount of sea salt aerosol in all the sites, as revealed by the snow concentration trend, but in sites with less snowflake falling we observe a lower deposition flux of sea salt aerosol.

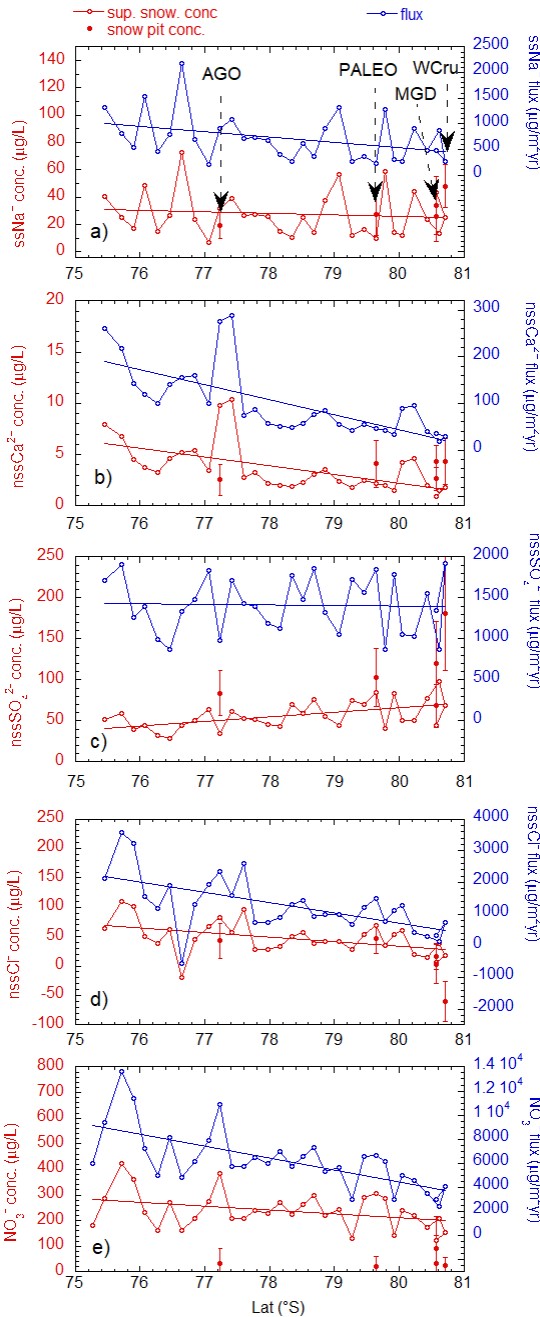

**Figure 6**. SsNa$^+$, nssCa$^{2+}$, nssSO$_4^{2-}$, nssCl$^-$ and NO$_3^-$ concentrations (red) and fluxes (blue) in the superficial snow samples (empty dots) and mean ± 1σ concentration in snow pit samples (red dots and vertical lines) as function of latitude (see Fig. 2 and Table 1S for the corresponding increases of distance from Dome C and distance from Indian Ocean). The linear fitting is reported in each plot to highlight the trend.



Crustal aerosol is found to be mainly dry deposited in the inner Antarctic plateau (Albani et al. 2012 and references therein) and, if the crustal aerosol supply to the plateau were constant, its deposition flux should increase moving South from Dome C,

due to a progressively lower accumulation rate. On the contrary, a decreasing trend of nssCa$^{2+}$ deposition fluxes as accumulation decreases is evident (Fig. 6b), implying that atmospheric concentration of crustal aerosol decreases along the route from Dome C as latitude increases.

Excluding the sporadic input from volcanic eruptions, the majority of nssSO$_4$$^{2-}$ found in the snow and aerosol of Antarctica comes from marine biological activity (e.g. Becagli et al. 2022; Kaufmann et al. 2010; Wolff et al. 2006).

Sulfate concentrations measured in this work are of same order of magnitude with respect to those reported for Dome Fuji (Iizuka et al., 2006), Dome C (Udisti et al., 2004) and in general for the Antarctic Plateau (Bertler et al., 2005), but an increasing trend is observed as latitude increases.

Legrand and Mayewski (1997) showed that in low accumulation sites in the inner Antarctic Plateau sulfate is deposited by dry deposition therefore change in concentration in snow layer should be ascribed to a variation in accumulation rate.

Indeed, as above reported a decreasing trend of accumulation rate was observed as the distance from Indian Ocean (and latitude) increases. Considering the sulphate flux it's clear that the amount of nssSO$_4$$^{2-}$ deposited is constant, indicating almost constant atmospheric concentrations in this area and confirming that dry deposition as main deposition process.

This evidence has an important implication as depositional flux of dry deposition is given by:

$$F = C_{atm} *v$$

Where $C_{atm}$ is the concentration of nssSO$_4$$^{2-}$ into the atmosphere and v is the deposition velocity. The deposition velocity depends on the size distribution of the considered species, in addition to the wind speed and relative humidity of the atmosphere. Deposition velocity spans between 0.03 to 0.3 cm/sec for the submicrometric particles (Duce et al. 1991). A

previous study of nssSO$_4$$^{2-}$ in size segregated aerosol at Dome C reveal that nssSO$_4$$^{2-}$ are mainly distributed in the submicrometric fraction showing an average annual concentration in this fraction of about 30 ng/m$^3$ (as average for the year from 2005 to 2012, Becagli et al., 2022). Considering this value as the mean annual aerosol concentration and a deposition velocity of 0.15 cm/sec we obtain a mean annual deposition flux of about 1400 μg/m$^2$yr that completely agrees with the value (1370 μg/m$^2$yr) calculated as average of the fluxes arising from nssSO$_4$$^{2-}$ concentration in the superficial snow collected at

each site of the traverse.

The flux variations around the average values are likely due to different mix of summer and winter snow precipitation collected in the most superficial snow. Indeed, considering the nssSO$_4$$^{2-}$ concentration measured in mid-summer (85 ng/m$^3$) and mid-winter (5 ng/m$^3$) in the aerosol submicrometric fraction at Dome C by Becagli et al. (2022), fluxes change during the year from about 4000 μg/m$^2$ yr to a minimum of 235 μg/m$^2$. This evidence demonstrates that atmospheric concentration of nssSO$_4$$^{2-}$ in

the plateau is spatially constant, but obviously it varies seasonally. Moreover, it has been shown that nssSO$_4$$^{2-}$ flux in EPICA-



Dome C ice core has not significantly changed throughout the last eight glacial cycles, which was interpreted to indicate a non-significant change in marine biogenic activity (Wolff et al., 2006). This conclusion is inconsistent with data derived from marine sediment cores that shows lower productivity at latitudes higher than 50°S during the last glacial period than during the current warm period (Kohfeld et al., 2005) that should reflect a lower flux of $nssSO_4^{2-}$ during glacial. The results achieved

by Ishino et al. (2019) by sulfur isotopic measurements may allow explaining this apparent controversial evidence. The Authors found a higher contribution of a non-biogenic source of $nssSO_4^{2-}$ contribution in glacial time than in interglacial, demonstrating that this source is consistent with sulfate crustal input. To estimate the crustal fractions of sulphate, the $nssCa^{2+}$ fluxes were multiplied by the $SO_4^{2-}/Ca^{2+}$ ratio in the uppermost Earth crust and (0.592 w/w, Bowen, 1979). The contribution of crustal source is about 10% at sites nearest to Dome C confirming the mean annual value obtained by Ishino et al. (2019) at present

day from aerosol measurements. It is interesting to notice that the contribution of the crustal source to $nssSO_4^{2-}$ decreases as latitude increases, making the sites far from Dome C more suitable to infer the past variation of the biogenic activity from $nssSO_4^{2-}$ fluxes.

As above reported, $nssCl^-$ arises from the exchange reactions between chloride in the sea salt aerosol and acidic species such as $H_2SO_4$ and $HNO_3$ (reaction 1 and 2). Therefore, the positive value of $nssCl^-$ indicates an extra source of HCl transported

and deposited separately respect to unmodified sea salt aerosol. Conversely, if the reaction occurs in the snow layers at the expense of the already deposited sea salt aerosol, $nssCl^-$ are negative. Therefore, the negative values of $nssCl^-$ represent the loss of chloride from snow layers in the form of HCl from $Cl^-$ deposited as NaCl and MgCl (i.e. arising form of sea spray). This is due to the reactions (1) and (2) that moves towards the formation of HCl due to its more volatile nature compared to $H_2SO_4$.

In superficial snow concentration and fluxes of $nssCl^-$ show a general decreasing trend with always positive values, demonstrating the presence of HCl in the freshly deposited snow. The decreasing trend as accumulation rate decreases is due to the wet and dry deposition processes both occurring for this species. By reporting the flux as function of accumulation rate an exponential function is found (Fig. 7) where the values for accumulation ratio = 0 is due to the dry deposition. Fig.7 shows that dry deposition accounts for a minor amount of this species being deposited mainly as wet deposition also at sites with low

accumulation rate. Another effect leading to a decreasing in $nssCl^-$ concentration as accumulation rate decreases is the post depositional loss of HCl, that can be highlighted by reporting the $nssCl^-$ concentration as function of depth (Fig. 8). Break point analysis was performed at the concentration vs. depth profile for each site aiming to find the depth at which post depositional loss ends and the concentration preserved at the end of reemission processes. A single o none break point was assumed to be present in each profile and its coordinates were computed by means of linear estimation using the "segmented"

package in R software (Muggeo, 2003, https://cran.r-project.org/web/packages/segmented/index.html). Therefore, the x value of break point coordinates (if found in the depth range of the snow-pit) represents the depth at which end the active layer (active for re-emission processes) and start the archive layer. The y coordinates the archived concentration at the site.



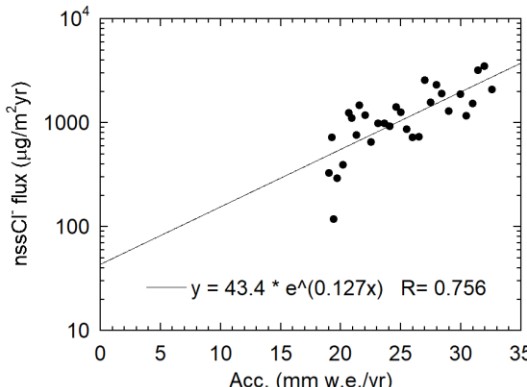


**Figure 7**. Scatter plot of nssCl⁻ flux vs accumulation rate. NssCl⁻ are reported in logarithmic scale, therefore the exponential fitting appears as linear in this plot.

In all the snow-pits a decreasing trend of nssCl⁻ concentration as depth increases is observed, but only at AGO5 the constant
concentrations are positive. Therefore, only at this site the deposited HCl partially stays stably fixed in the deep layer. Considering an average value of 100 μg/L of HCl deposited in the superficial layer at AGO5, only the 24% is preserved at depth higher than 1m. At PALEO, MGD-A and MGD-E the decreasing trend of nssCl⁻ do not reach stable values in the depth interval covered by the snow-pit, and negative values (evidencing the chloride depletion of the deposited sea salt aerosol) are reached at depths that are more and more shallow as accumulation rate decreases. A particular trend is visible at WCru (where
the negative values are already reached at 10 cm depth and constant negative concentrations of about -70 μg/L are reached at about 60 cm depth. The loss of HCl from sea spray in the most superficial 10 cm is due to the presence of ice already at the surface indicating an older layer than expected considering the snow accumulation layer or rapid ice formation due to the wind scouring. Both processes result to a loss of HCl.

Nitrate is present as acidic species in the atmosphere over the Antarctic Plateau. It is produced by the oxidation of nitrogen
oxides ($NO_x$ -> $NO + NO_2$) coming from the downward transport of $NO_x$ produced by stratospheric sources, the long-range transport of air masses enriched in $NO_x$ by tropical lightning (Legrand and Kirchner, 1990), sedimentation of polar stratospheric clouds (Tritscher et al. 2021), and from the action of the snowpack, which can act both as a source and as an accumulator of $HNO_3$ and its precursors (Savarino et al. 2007). High concentrations and fluxes are measured in all the superficial snow samples collected along the traverse (fig. 6). Besides, at WCru, MGD-a and MGD-E sites concentration in the first sample of snow pits
(representing the most superficial 3 cm layer) are higher than the concentration measured in the superficial snow (representing the most superficial 10 cm layer) (Table 3).







**Figure 8.** Nitrate and nssCl⁻ concentrations as function of depth in the snow pits dug at AGO-5, PALEO, WCru, MGD-A, and
MGD-E . The two linear regressions obtained by the break point analysis are plotted for each site, the coordinates of the lines
intersection are reported in each plot (x; y). For nssCl⁻ at PALEO, MGD-A and MGD-E the break point analysis has not
statistical significance and the regression lines fitting the data points are not reported. The dashed line at 0 value for nssCl⁻
highlight the presence of HCl (for nssCl⁻>0) and the Chloride depletion from the already deposited sea salt aerosol (nssCl⁻<0).



HNO$_3$ is deposited by scavenging during formation of precipitation, uptake of HNO$_3$ onto the ice crystal's surface during and after precipitation and by dry deposition (Röthlisberger et al., 2002). It was demonstrated that in the case of liquid or mixed clouds, essentially all HNO$_3$ is removed from the gas phase independent of the cloud temperature (Seinfeld and Pandis, 1998), conversely the co-condensation of HNO$_3$ and water molecules on ice crystals would lead to a very low bulk concentration in the fresh snow (Thibert and Dominé, 1998). Therefore, in the absence of liquid water, i.e. in ice clouds, the high NO$_3^-$

concentrations found in surface snow could not be explained. The main processes leading to the high concentration in snow surface are the dry deposition and the uptake on snow crystals. The dry deposition was calculated by the mean atmospheric HNO$_3$ measured in December-January at Dome C (Erbland et al., 2013) of about 100 ng/m$^3$ and assuming it as constant in this area. Literature data report for HNO$_3$ different values for the dry deposition velocity; for instance, at South Pole station and DML (which have a similar snow accumulation rate) a value of 0.8 cm/s was used (Huey et al., 2004; Winton et al., 2020).

Other estimates of dry deposition velocities include 0.05– 0.5 cm/s for HNO$_3$ over snow (Hauglustaine et al., 1994; Seinfeld and Pandis, 1998) and a higher value for areas over the open ocean (1.0 cm/s, Duce et al., 1991) due to the size distribution shifted toward high diameters in this environment. Finally, a deposition velocity of 0.15 cm/s was used for summer HNO$_3$ at Dome C (Erbland et al., 2013). The estimated NO$_3^-$ deposition velocity at Dome C is low because of the strong recycling of NO$_3^-$ on the polar plateau in summer; i.e. reactive nitrogen is re-emitted from the skin layer to the atmosphere. We assume that

the value used by Erbland et al. (2013) could better reflect the deposition velocity in this area, and the value of 0.15 cm/s was used to calculate the concentration and fluxes due to this process at each site (Table 3). Table 3 shows that the calculated concentration expected considering only the dry deposition are lower than the measured. This evidence can be due to (i) considering constant the concentration of NO$_3^-$ into the atmosphere and a low deposition velocity; (ii) other deposition processes (snow uptake) being more efficient as distance from Dome C increases; (ii) post depositional processes occurring in

most superficial centimeters of snow.

It is not possible to verify the first issue without measurement of HNO$_3$ into the atmosphere, but in case the species is only dry deposited, the trend of snow concentration should show an increase as accumulation rate decreases. Conversely Fig.6e shows a decreasing trend in concentration, therefore the dry deposition cannot be the only deposition process. The adsorption of HNO$_3$ on ice surfaces is temperature dependent with higher uptake at lower temperatures (Abbatt, 1997; Jones et al., 2014).

The surface uptake of HNO$_3$ is estimated using a linear regression through the values for temperature dependent uptake found by Abbatt (1997) and assuming a typical surface area of 4000 m$^2$/m$^3$ (Narita,1971). The temperature data (daily maximum and minimum) are from automated weather stations installed in the Megadune area and at Paleo site during the traverse and derived by model for the other sites (Casado et al., 2022). Temperature varies daily following maxima and minima of solar radiation, but there are not large differences among the sites during the time of the traverse (Casado et al., 2022). The calculated

concentrations by the snow uptake (table 3) are generally higher than the measured concentrations (also considering the higher measured temperatures at each site determining a lower uptake), but they reflect the same trend as distance from Dome C increases. Indeed, at Dome C, AGO5 and PALEO the concentration decreasing trend can be explained by the decreasing temperature at the beginning of the traverse, lower temperature favoring the uptake by superficial snow. Going forward with



the traverse, min and max temperature remain constant, but as density of snow is decreases, decreases the volume of water

containing the surface area of uptake. Indeed, the lower calculated concentration are at WCru and MGD-E where superficial layer has a higher density due to the ice crystal formation. In the two megadune sites (same temperature, but different superficial snow density) the measured $HNO_3$ concentration is higher at MGD-A site with lower snow density. Therefore, the $HNO_3$ deposition in summer is a combination of dry deposition and snow uptake, the latter having the potentiality to give highest concentration in superficial snow.

In winter when temperature reach -80°C, the surface uptake is more active, but at this time $HNO_3$ into the atmosphere is low (Erbland et al., 2013; Traversi et al., 2014) therefore the main deposition occurs in summer. It is known that $HNO_3$ is not stably fixed in the snow and post depositional effect are well documented in sites with low accumulation rate (Savarino et al.,2007; Traversi et al., 2009;  Erbland et al., 2013; Winton et al., 2020, Akers et al., 2022). Post depositional processes occurs in the first centimeters of snow, isotopic measurements performed at Dome C revealed that in the central Antarctic plateau in summer

the dominant reemission process is photolysis (Erbland et al.,2013). The reemission process occurring in winter when the absence of UV radiation does not allow the photolysis of $HNO_3$ are less clear and less documented, but as above reported the deposition in wintertime is low.

Reporting the $NO_3^-$ concentration as function of depth in the 5 snow pits it is possible to quantify the impact of post deposition processes in term of rate of loss and $NO_3^-$ concentration preserved in the snow (Fig. 8).

Fig. 8 shows that the re-emission of $HNO_3$ occurs fast in the most superficial centimetres of snow in all the sampling sites and it is concluded in 20 cm of deposition at maximum. The background $NO_3^-$ concentrations preserved in the snow layer decrease as the accumulation rate decreases. An apparent exception to this is represented by the pattern observed at MGD-A where the break point concentration is 111 µg/L (i.e higher than at sites with higher accumulation rate). In this case it has to be noticed that this concentration is not constant but further decreases as depth increases. Therefore, in this case the actual $NO_3^-$

concentration preserved at this site is lower than this value and it is not detectable in the depth range covered by the snow pit. Particular attention has to be paid to the comparison between the two sites in the megadune area: the accumulation site (MGD-A) presents higher concentration at the surface than MGD-E likely due to the relative position  respect to katabatic wind favouring the snow redistribution at MGD-E, besides the active layer seems to be the same (about 12 cm of snow) but below this layer at MGD-A the re-emission of $HNO_3$ is still active, even if less efficient. Another interesting feature is that sites

affected by wind erosion (MGD-E) and formation of ice crystals (WCru) present still efficient reemission processes with the lowest $NO_3^-$ concentration (about 8.5 µg/L) preserved in the snow layers (Fig. 8 e Tab. 3).






**Table 3.** Nitrate concentration in superficial and sub-superficial snow layers considered as the concentration stably fixed in several sites in the Antarctic Plateau.

*Accumulation rate for WCru, MGD are evaluated from the regression line as reported in Fig. 9. The sites MGD-A and MGD-
E are at the same distance from the Indian Ocean therefore only one accumulation rate is calculated. This value represents an intermediate value between the two sites sampled in magadune area having different accumulation rate. †The reported accumulation rate for Dome C are from 1992-2016 reported in (Caiazzo et al., 2021).

| Reference | Sampling point | Acc.rate (mm w.e./yr) | Temp. (Max) | Density (kg/L) | Conc. Snow uptake | Conc. Dry dep | Concentration in superficial snow - µg/L (depth of considered layer - cm) | Concentration stably fixed (µg/L) (depth of considered layer - cm) |
|---|---|---|---|---|---|---|---|---|
| This study | AGO-5 | 25.7 | -25 | 0.28 | 1106 | 184 | 307 (0-3cm); 385 (0-10cm) | 30.5 (22.0 cm) |
| This study | PALEO | 22.6 | -20 | 0.26 | 757 | 209 | 270 (0-3cm); 304 (0-10cm) | 17.5 (12.6cm) |
| This study | WCru | 19.0* | -20 | 0.37 | 532 | 249 | 188 (0-3cm); 155 (0-10cm) | 8.7 (18.6cm) |
| This study | MGD-A | 19.4* | -20 | 0.28 | 703 | 244 | 793 (0-3cm); 174 (0-10cm) | - |
| This study | MGD-E | | -20 | 0.35 | 562 | 244 | 607 (0-3cm); 207 (0-10cm) | 8.4 (12.6 cm) |
| Shi et al., 2018 | Dome A | | | | | | 350 (0-3cm) | |
| Udisti et al., 2004 | Dome C | 35.3 † | -25 | 0.3 | 1108 | 134 | 381 (0-2cm) | |
| Traversi et al., 2017 | Dome C | 35.3 † | 0.3 | 1108 | 134 | 0.3 | 200 (0-2cm); 800 (0-0.5cm) | |
| Erbland et al., 2013 | Dome C | 35.3 † | 0.3 | 1108 | 134 | 0.3 | 100 / 600 (0-10cm) | |
| Iizuka et al., 2006 | Dome Fuji | | | | | | | 40.3 (0-3.4 m) |


Given the different efficiency of post-depositional processes and the dependence of $NO_3^-$ concentration on accumulation, time of year and depth of the sampled superficial snow layer, a direct comparison with data from other sites is not possible. Table 3 shows data from several studies carried out on snow samples of specified thickness collected in summer and concentration stability fixed in snow layer for each site.




## 5-Conclusions

This paper reports the data of chemical composition and spatial distribution of aerosol chemical markers for unexplored region of the Antarctic Plateau along the EAIIST traverse from Dome C towards the South Pole. The study of the spatial variability
of chemical impurities in the snow allows to understand the deposition and post deposition processes of aerosol and gaseous species in sites with very low snow accumulation rate and especially in sites characterized by particular glaciological characteristics such as megadune and wind crust.

The $nssSO_4^{2-}$ arising from biogenic emission is the dominant source of background sulphate, but the volcanic signature of Pinatubo eruption occurred in 1991 and visible at AGO-5 and PALEO allow the determination of accumulation rates that are
25.7 and 22.6 mm of water equivalent/year respectively at the two sites.

The insoluble dust profiles measured at AGO5 and PALEO are both presenting a peak at 61.5-64.5 cm and 37.5-40.5 cm respectively. These peaks are characterised by an increase in the concentration of total dust particle with a mineral dust grain size progressively decreasing. The stratigraphic location of the peak is compatible with deposition of volcanic ash from the Puyehue-Cordón Caulle explosive eruption event of June 2011 also detected in West Antarctica.

Considering the accumulation rate at Dome C (Caiazzo et al., 2021; Traversi et al., 2009)  and those obtained from AGO-5 and PALEO snow-pits it is possible to highlight a decreasing trend of accumulation rate as the distance from the Indian Ocean increases and considering linear the decreasing trend the accumulation rate the accumulation rate and deposition fluxes for the main markers in each superficial snow sampling site was calculated.

The $ssNa^+$ accounting for the 92.5% of the total $Na^+$ was chosen as marker of sea spray deposition and it is preserved stably in
the snow layers. SsNa shows a constant trend in concentrations (around 30 $\mu g/m^3$) and a decreasing trend in fluxes as latitude, distance from Dome C and distance to the Indian Ocean increase. This pattern demonstrates that, despite the very low accumulation rate in this area, the main deposition process of sea spray aerosol is the wet deposition.

Conversely, $nssSO_4^{2-}$ sulfate concentration presents an increasing trend as accumulation rate decreases, indeed by calculating the flux of $nssSO_4^{2-}$ it results to be constant along the traverse, but the relative importance of the fluxes of biogenic and crustal
sources for $nssSO_4^{2-}$ both dry deposited, change as distance from Dome C increases. This implies that constant atmospheric concentrations of $nssSO_4^{2-}$ in this area arise from different contributions of the two sources with the impact of biogenic source that increases as distance from Dome C increases. The presence and quantification (by $nssCa^{2+}$) of a dry deposited crustal source as the biogenic one provides a further evidence of previous findings based on isotopic measurement of S in the $nssSO_4^{2-}$ (Ishino et al., 2019) and sheds light on the paradox of constant biogenic activity during glacial and interglacial time as Antarctic
ice core seems to show (Wolff et al., 2006), on the contrary of marine sediment cores (Kohfeld et al., 2005). Indeed, the measured constant fluxes of $nssSO_4^{2-}$ in glacial and interglacial periods do not mean that biogenic source (and therefore the biogenic activity in the Antarctic Ocean) is constant; in fact, during the glacial time the crustal source of $nssSO_4^{2-}$ increases, meaning that the biogenic $nssSO_4^{2-}$ decreases, thus indicating a reduced biogenic productivity respect to interglacial time.





NssCl$^-$ represent the fraction of Cl$^-$ deposited as HCl and arises from the exchange reactions between chloride in the sea salt
aerosol and acidic species such as $H_2SO_4$ and $HNO_3$. If these reactions occur in the atmosphere, HCl is transported and
deposited separately respect to unmodified sea salt aerosol. HCl is deposited by wet deposition but it undergoes post
depositional processes that are more evident as accumulation rate is lower and especially in sites affected by wind crust
formation. In these sites a Chloride depletion (negative value of nssCl$^-$) is already shown in the first 10 cm of snow. Cl$^-$
depletion is due to the reemission into the atmosphere of HCl deposited as sea salt by the exchange reaction with $H_2SO_4$.

Another important marker in ice core is $HNO_3$, it is deposited mainly in summer both by dry deposition and especially by
superficial snow uptake (the lower is the temperature the higher is the uptake), but it is reversibly deposited. Previous study
revealed that the main reemission process in summer is $HNO_3$ photolysis (Erbland et al., 2013), but snow redistribution
processes seem to have a role in determining the concentration in the superficial snow. Indeed, by comparing accumulation
and ablation sites at MGD the highest concentration on the most superficial 3 cm of snow is measured in the accumulation site
(MGD-A). Besides, among all the sites the lowest values in the superficial snow layer are measured at WCru, the site most
affected by wind scouring.

Break point analysis applied on $HNO_3$ concentration vs depth profile allow to establish the depth of the active layer and the
concentration of $NO_3^-$ stably preserved in the snow in each site. The depth of the active layer is very low and span from 22 cm
to 12 cm and concentration preserved in the snow decreases as the accumulation rate decreases, the lower concentration is
preserved at WCru and MGD-E (8.5 µg/L) showing again that wind scouring play a role in the re-emission processes in the
active layer.

The knowledge and quantification of all the above reported processes will allow the interpretation of the ice core stratigraphies
as function of environmental and climatic variation in the past. Besides, some processes that are visible in the shallow snow
layers at sites characterized for instance by ice crust formation, can be considered as analogous of those occurring in the deepest
snow layers. The knowledge of such processes may allow interpreting the ice core stratigraphy in low accumulation site likely
recording, at selected sites, the climate history of more than one million years ago (Wolff et al., 2022).

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

**Data availability**

Data set is available on PANGAEA at https://issues.pangaea.de/browse/PDI-34759 and on request at the corresponding author.

**Founding**

This study was founded by the ANR EAIIST (grant ANR-16-CE01-0011), of the French Agence Nationale de la Recherche,
the BNP-Paribas foundation and its Climate Initiative Program, the Institut Polaire Français IPEV, the MIUR (Ministry of

Education, University and Research) through the PNRA (National Antarctic Research Program) EAIIST project (grant
PNRA16_00049-B) and PRIN2017 project "Innovative Analytical Methods to study biogenic and anthropogenic proxies in
Ice COres" AMICO (grant PRIN_2017EZNJWN_006).

**Author contributions.**

SiB, RT and MB conceived and conceptualized the idea. The IC measurements were carried out by SV, SaB and RN with the
supervision of MS. Dust measurements were carried out by CA, BD. The samples were collected by AS, JS. Sampling site
strategy was developed by MF and JS. All authors contributed to the data analysis and interpretation of the results. SV, SaB
and SiB wrote the paper, using feedback from all other co-authors.

**Acknowledgements**

The authors would like to warmly thank all the participants of the EAIIST traverses for their hard field contributions allowing
the collection of the crucial in-situ measurements used in this study.
