# Peer review of "Spatial and temporal variations in surface snow chemistry along a traverse from Dome C toward South Pole in the framework of East Antarctic International Ice Sheet Traverse (EAIIST) project."

_EGUsphere, 2023_

## Referee Comment (RC2)

Comments on Ventisette et al., entitled "Spatial and temporal variations in surface snow chemistry along a traverse from Dome C toward South Pole in the framework of East Antarctic International Ice Sheet Traverse (EAIIST) project"

Ventisette et al. reported the major chemical ion concentrations (eight species) in surface snow and snow pits collected along an inland East Antarctic traverse route from Dome C towards South Pole. The authors are to be congratulated on sampling the valuable samples on the East Antarctic plateau where has not been explored yet, and the samples may be usually sampled under the very hash conditions. The data provided in this manuscript will be of significance towards a better understanding of the behaviors of the chemicals (eg, the deposition and post-depositional processes) and the interpretation of ice core records on East Antarctic plateau. In this case, the data, in my opinion, deserve publication in Cryosphere. In general, the main findings of this study are similar to previous investigations on the East Antarctic plateau (eg, Dome A, Dome C, and Dome F) and in fact no new/innovative scientific findings are present in this work, but this does not conceal the value of the data. In my opinion, some of the data are over-interpreted, and sometimes are incorrectly interpreted (see the comments below). I suggest the authors focus on their new data (e.g., non sea salt fractions of the ions and the dust concentration) and make a comprehensive comparison with previous observations on other Antarctic plateau sites. Also, the paper may be significantly shortened, like the "Brief Communication" rather than a "Full Research Article".

1) The current version includes too much general description or discussion that is well-known. For instance, the mechanisms of the occurrence of $nssCl^-$ and the transport of HCl; the main sources of $nssSO_4^{2-}$ and the snow pit dating using the spikes of $nssSO_4^{2-}$, and these points are generally well-known, and the authors may not have to spend a lot of space to discuss these. The current version of the manuscript then can be shortened.

2) My biggest concern is the estimation of the snow accumulation rate and the chemical flux at different sampling sites. The authors have only three data points, Dome C, AGO-5, and PALEO, along the large traverse route which extends about 600km, then they established a linear function between the snow accumulation rate and the distance from Indian Ocean to estimate the snow accumulation rate at each surface snow sampling sites (about 30 sites). This approach, in my opinion,

is not reasonable. Firstly, the available data are very limited, and it is difficult to establish a precise relationship between accumulation and the distance to the coast. Secondly, the snow accumulation rate may vary significantly among sites (wind crust area and the snow accumulated area), and it will be influenced by a variety of factors (wind, slope, aspect, etc.) in addition to the distance from coast. Therefore, the estimation of snow accumulation rate based on one single factor (distance from coast) may suffer from large uncertainty. The authors may remove the sections on the snow accumulation and ion flux estimation, to re-interpret their data.

3) The discussion of nitrate in snow may be significantly shortened. A number of investigations have been done across East Antarctic plateau, although the knowledge on snow nitrate in the region of this study is unavailable. Previous studies tried to understand the deposition and post-depositional processes of nitrate using the surface (skin layer) snow, snow pits, and the atmosphere samples, with the aid of the isotopes of nitrate and the modeling. The current study, without the atmosphere samples, mainly interpret their observed nitrate concentration data based on the previous work, and in my opinion, no new knowledge/information on snow nitrate behaviors is present in the current version. In addition, this study does not provide the data of the atmosphere, thus the calculation of the atmospheric deposition flux of nitrate and the non-sea-salt sulfate ($F = C_{atm} * v$) remains uncertain. Thus, the current version of the manuscript could be shortened.

Specific comments

1. The current title is too long, may be "Spatial and temporal variations in surface snow chemistry along a traverse from Dome C toward South Pole"

2. L32, should be "$Ca^{2+}$"

3. L55, should be Qin et al., 1992, rather than Dahe et al., 1992.

4. L59-61, could the authors briefly introduce the scientific aims of the EAIIST project?

5. Figure 1, the geographic "South Pole" should be included in the figure, and the coordinate system unit can be changed to "decimal degrees"?

6. In sampling method section, the authors will have to clarify what the "top ~6cm snow" can present? In my opinion, may be at least one-year of snow accumulation? It is noted that chemical ion concentrations may vary significantly

among snow depths. In addition, I cannot find the surface snow density data in the manuscript.

7. L123-125, eight species of chemical ion were determined, but I noticed that some of the data are not included in the paper. I suggest all of the data be included, for the reference of the future research.

8. L134, typos of the chemical formula.

9. L149-150, how did the authors get the reproducibility? I donot think the unit of the detection limit in Table 1 is correct, mg $L^{-1}$? For example, 3.0mg/L for nitrate, that is, 3000ppb?

10. L155, more details on the determination of dust in the snow should be present, and the d.l.?. All of the snow pit samples were analyzed for the particle concentrations?

11. L167, should be "$Na^+$" rather than "Na", similar for Ca in L174

12. L199-204, I generally agree with the authors that most of the sulfate is from the secondary sources, but the sea salt aerosols from sea ice cannot reach the Antarctic plateau even in winter?

13. Section 4.2, this section may be significantly shortened. In addition, I find that the concentration peaks of dust are corresponding to the valleys of the non sea salt sulfate at PALEO in Figure 4. Thus, I am not sure whether this layer corresponds to the Puyehue-Cordón Caulle (Chile) explosive eruption event of June 2011. From the dating results of the dust concentration, what is the snow accumulation rate at the two sites? Are they comparable to those inferred from the Pinatubo eruption signals? The dust concentration data at the base of the snow pit are not available (different from those of the non sea salt sulfate)?

14. L265-265, I don't think it is reasonable to estimate the snow accumulation rate at each sampling site using the relationship of distance versus accumulation in Table 2. Also see my general comments.

15. Section 4.3, please see my general comments, the author may will re-organize this part.

16. The Conclusion section may be significantly shortened, following the main text.

---

## Author Comment (AC2)

**Answer to the anonymous referee 2**

Ventisette et al. reported the major chemical ion concentrations (eight species) in surface snow and snow pits collected along an inland East Antarctic traverse route from Dome C towards South Pole. The authors are to be congratulated on sampling the valuable samples on the East Antarctic plateau where has not been explored yet, and the samples may be usually sampled under the very hash conditions. The data provided in this manuscript will be of significance towards a better understanding of the behaviors of the chemicals (eg, the deposition and post-depositional processes) and the interpretation of ice core records on East Antarctic plateau. In this case, the data, in my opinion, deserve publication in Cryosphere. In general, the main findings of this study are similar to previous investigations on the East Antarctic plateau (eg, Dome A, Dome C, and Dome F) and in fact no new/innovative scientific findings are present in this work, but this does not conceal the value of the data.

**Thank for supporting this work!**

In my opinion, some of the data are over-interpreted, and sometimes are incorrectly interpreted (see the comments below). I suggest the authors focus on their new data (e.g., non sea salt fractions of the ions and the dust concentration) and make a comprehensive comparison with previous observations on other Antarctic plateau sites. Also, the paper may be significantly shortened, like the "Brief Communication" rather than a "Full Research Article".

**I'm not sure that the paper, even shortened, has the characteristics of "Brief Communication", but if the editor agrees, that's fine with me. In any case, the work can be shortened by following the advice of both referees and remain a "Full Research Article".**

1) The current version includes too much general description or discussion that is well-known. For instance, the mechanisms of the occurrence of nssCl- and the transport of HCl; the main sources of nssSO4 2- and the snow pit dating using the spikes of nssSO4 2- , and these points are generally well-known, and the authors may not have to spend a lot of space to discuss these. The current version of the manuscript then can be shortened.

**The referee is right I can short this part.**

2) My biggest concern is the estimation of the snow accumulation rate and the chemical flux at different sampling sites. The authors have only three data points, Dome C, AGO-5, and PALEO, along the large traverse route which extends about 600km, then they established a linear function between the snow accumulation rate and the distance from Indian Ocean to estimate the snow accumulation rate at each surface snow sampling sites (about 30 sites). This approach, in my opinion, is not reasonable. Firstly, the available data are very limited, and it is difficult to establish a precise relationship between accumulation and the distance to the coast. Secondly, the snow accumulation rate may vary significantly among sites (wind crust area and the snow accumulated area), and it will be influenced by a variety of factors (wind, slope, aspect, etc.) in addition to the distance from coast. Therefore, the estimation of snow accumulation rate based on one single factor (distance from coast) may suffer from large uncertainty. The authors may remove the sections on the snow accumulation and ion flux estimation, to re-interpret their data.

**Both referees agree with the improper evaluation of the accumulation trend along the traverse. I also agree that is a simplify approach and the large variability observed in the fluxes around the linear trend demonstrate that, other than distance from the sea, several factors affect the accumulation rate. The similarity in slope (but opposite sign) between accumulation rate and nssSO4 concentration resulting in constant fluxes of nssSO4 (confirming the general knowledge that dry deposited), had encouraged me to assume generally correct this approach. Anyway, as both referees disagree, I can delete this part of discussion.**

3) The discussion of nitrate in snow may be significantly shortened. A number of investigations have been done across East Antarctic plateau, although the knowledge on snow nitrate in the region of this study is unavailable. Previous studies tried to understand the deposition and post-depositional processes of nitrate using the surface (skin layer) snow, snow pits, and the atmosphere samples, with the aid of the isotopes of nitrate and the modeling. The current study, without the atmosphere samples, mainly interpret their observed nitrate concentration data based on the previous work, and in my opinion, no new knowledge/information on snow nitrate behaviors is present in the current version. In addition, this study does not provide the data of the atmosphere, thus the calculation of the atmospheric deposition flux of nitrate and the non-sea-salt sulfate (F = Catm *v) remains uncertain.

**The referee is right, without isotopic signature and assuming incorrect the accumulation rate, main part of the discussion here is only speculative. Also in this case, I can short this part.**

Specific comments

1. The current title is too long, may be "Spatial and temporal variations in surface snow chemistry along a traverse from Dome C toward South Pole"

**Yes, this can be done**

2. L32, should be "Ca2+"

**Correct**

3. L55, should be Qin et al., 1992, rather than Dahe et al., 1992.

**Correct**

4. L59-61, could the authors briefly introduce the scientific aims of the EAIIST project?

**Yes**

5. Figure 1, the geographic "South Pole" should be included in the figure, and the coordinate system unit can be changed to "decimal degrees"?

**Yes**

6. In sampling method section, the authors will have to clarify what the "top ~6cm snow" can present? In my opinion, may be at least one-year of snow accumulation? It is noted that chemical ion concentrations may vary significantly among snow depths. In addition, I cannot find the surface snow density data in the manuscript.

**Yes, the plan was to sample one year of deposition. Snow density will be added in table 1S**

7. L123-125, eight species of chemical ion were determined, but I noticed that some of the data are not included in the paper. I suggest all of the data be included, for the reference of the future research.

**The referee is right but the majority data on NH4 are below the detection limit.**

8. L134, typos of the chemical formula.

**Correct**

9. L149-150, how did the authors get the reproducibility? I donot think the unit of the detection limit in Table 1 is correct, mg L-1 ? For example, 3.0mg/L for nitrate, that is, 3000ppb?

**Yes, this is a mistake, in the formatting process I lost the symbol character, μg/L or ppb is the correct measuring unit.**

10. L155, more details on the determination of dust in the snow should be present, and the d.l.?. All of the snow pit samples were analyzed for the particle concentrations?

**Detail on this can be add.**

11. L167, should be "Na+ " rather than "Na", similar for Ca in L174

**Yes, correct.**

12. L199-204, I generally agree with the authors that most of the sulfate is from the secondary sources, but the sea salt aerosols from sea ice cannot reach the Antarctic plateau even in winter?

**Correct, more detail will be report here.**

13. Section 4.2, this section may be significantly shortened. In addition, I find that the concentration peaks of dust are corresponding to the valleys of the non sea salt sulfate at PALEO in Figure 4. Thus, I am not sure whether this layer corresponds to the Puyehue-Cordón Caulle (Chile) explosive eruption event of June 2011. From the dating results of the dust concentration, what is the snow accumulation rate at the two sites? Are they comparable to those inferred from the Pinatubo eruption signals? The dust concentration data at the base of the snow pit are not available (different from those of the non sea salt sulfate)?

**As reported in the text at line 251, glass geochemistry is necessary to unequivocally infer the volcanic source. Anyway, the accumulation rate calculated by considering this signature as Puyehue-Cordón Caulle (Chile) explosive eruption and Pinatubo eruption are comparable. In the new version of the paper, if it will be accepted, a more detailed discussion on accumulation rate variability will be add.**

14. L265-265, I don't think it is reasonable to estimate the snow accumulation rate at each sampling site using the relationship of distance versus accumulation in Table 2. Also see my general comments.

**Ok, this part will be deleted.**

15. Section 4.3, please see my general comments, the author may will re-organize this part.

**This part will be strongly revised by deleting main part of discussion.**

16. The Conclusion section may be significantly shortened, following the main text.

**Conclusion will be change accordingly.**